# Assessment of antidiabetic, hepatoprotective, and analgesic effects of quinazolinone derivative, (E)-1-Benzoyl-3-((4- (Dimethylamino) Benzylidene) Amino)-2-(4-(Dimethylamino) Phenyl)-2,3 dihydroquinazoline-4(1h)-one, in diabetes induced mice model

Haider Ali[1], Asif Jan[1,2,3]*, Gowhar Ali[1], Syed Shaukat Ali[4], Abdur Rahim[5], Ursula Abu Nahla[6]*

1 Department of Pharmacy, University of Peshawar, Peshawar, Pakistan, 2 Saidu Group of Teaching Hospitals (SGTH), Saidu Sharif, Swat, Pakistan, 3 District Headquarter Hospital Charsadda, Charsadda, Pakistan, 4 Abdul Wali Khan University Mardan, Mardan, Pakistan, 5 CECOS University of Science and Technology Peshawar, Peshawar, Pakistan, 6 Faculty of Medicine, Hebron University, Hebron, Palestine

* uabunahla@gmail.com (UAN) asif.research1@gmail.com (AJ)

## Abstract

Diabetes can cause serious complications such as liver damage and nerve pain. Unfortunately, existing treatment options for these problems often have limited effectiveness and unwanted side effects. To find better therapeutic alternatives with good efficacy and safety profile this study tested a novel quinazolinone derivative, *(E)-1-benzoyl-3-((4(dimethylamino)benzylidene)amino)-2-(4-(dimethylamino) phenyl)-2,3 dihydroquinazoline-4(1H)-one*, in diabetic mice's experiencing liver damage and neuropathic pain. Diabetes was induced in mice using alloxan (150 mg/kg). For possible antidiabetic effect, test compound was given in doses of 10 mg/kg and 20 mg/kg. The standard drugs used for comparison was Glibenclamide (5 mg/kg), Tramadol (50 mg/kg) and Diclofenac sodium (50 mg/kg) for pain relief, and Gabapentin (75 mg/kg) for nerve pain. Pain relieving effect was assessed using various test models (e.g., hot plate, writhing, allodynia, and hyperalgesia). Liver function was studied through blood tests and tissue examination. The test compound (at test dose of 20 mg/kg) led to a significant reduction in blood glucose even greater than the reduction seen with glibenclamide (5 mg/kg). Similarly, the test compound significantly reduced pain and showed protective effects on the liver. This new quinazolinone compound was found to be safe and effective in reducing diabetic nerve pain and liver damage in mice. It may offer a better alternative to currently available treatments like gabapentin and glibenclamide.

**Data availability statement:** All relevant data are within the paper.

**Funding:** The author(s) received no specific funding for this work.

**Competing interests:** The authors declare that they have no known competing financial interests or personal relationships that could have influenced the work reported in this paper.

## Introduction

Diabetes is a global health issue that disrupts insulin function, raising infection risks, coma, macro and microvascular problems and even death [1]. Diabetes's most significant complication is nerve damage (neuropathy). Long-term diabetes can cause painful nerve sensations like allodynia, which is a sensation of pain from light touch, and hyperalgesia, which is increased pain to stimuli [2]. 15% to 25% of cases of painful diabetic neuropathy are caused by deterioration of the peripheral somato-sensory system. Neuropathy dramatically reduces patients' quality of life, impacting work, movement, emotions, sleep, and social life [3]. Research statistics indicate that about 25% to 50% of individuals with diabetes will experience symptoms of neuralgia, also known as painful diabetic neuropathy and that approximately half of those with diabetes will eventually develop diabetic peripheral neuropathy, which presents as abnormal sensations in the distal extremities.

Research in the literature has shown that long-term diabetes mellitus (D.M.) can cause a variety of disorders that impair the metabolism of fat and glycogen, including liver disorders like non-alcoholic fatty liver disease (NAFLD), abnormal glycogen accumulation, hepatocellular carcinomas (HCCs), fibrosis, cirrhosis, acute liver disease, viral hepatitis, and abnormally elevated hepatic enzymes [4]. Given the aforementioned research, it is evident that new, safe treatments are required for medication-induced neuropathic pain. This direction is being pursued with the current investigation.

Current research perspectives was kept on the assessment of quinazolinone compound against diabetic neuropathy concomitant with hepatic problems. Chemically, quinazolinones are heterocyclic aromatic molecules that contain nitrogen [5] formed when two nearby carbon atoms fuse a pyrimidinone ring with a benzene ring. Quinazolinones have been associated with several pharmacological properties, including anti-inflammatory, analgesic, antidepressant, antibacterial, antiviral, antifungal, antitubercular, antimalarial, antiproliferative, antileishmanial, antihypertensive, anti-diabetic, diuretic, antipsychotic, antiarrhythmic, anticonvulsive, and anticancer effects [6]. Based on the initial research and the previously established pharmacological effects of quinazolinones, a derivate of quinazolinone is chosen and assessed pre-clinically against diabetic neuropathy pain model. The derivative is named as *(E)-1-benzoyl-3-((4-(dimethylamino) benzylidene) amino)-2-(4-(dimethylamino) phenyl)-2,3 dihydroquinazoline-4(1H)-one*.

Elevated blood sugar levels trigger oxidative damage and the polyol pathway. An increase in sorbitol and fructose through the polyol pathway causes abnormalities in protein kinase C and Na+/K+ATPase [7], causing sodium retention, oedema, axoglial disjunction, myelin swelling, nerve degeneration, and reduced nerve function [8,9]. Similarly Amadori products are glycated residues produced when high glucose levels react non-enzymatically with the primary amino groups of proteins. The Amadori products undergo a sequence of fragmentation and dehydration processes that result in the formation of stable covalent adducts known as AGEs [10]. The pathophysiology of diabetes-related problems is mainly influenced by these non-enzymatic glycation and Advanced Glycation End Products (AGEs) [7].

Anti-phospholipid antibodies (PLAs) and anti-GM1 ganglioside antibodies, which target the body's own cell phospholipids and GM1, respectively, can also cause diabetic neuropathy. When either of these antibodies is activated due to an immunological condition, nerve issues may result. In PLAs, vascular thrombosis causes neurons to lose their blood supply [8].

Microvascular insufficiency has been linked to diabetic neuropathy. Studies reveal that abnormalities in the endoneurial or epineurial blood arteries, low PO2, increased vascular resistance, altered vascular permeability, and reduced neural blood flow can induce partial or complete ischemia in the nerves of diabetics [8].

One of the common and severely disabling microvascular consequences of diabetes mellitus that affects a large percentage of people worldwide is diabetic neuropathy. It is hypothesized that growth factor deficiency has a role in the genesis of diabetic neuropathy. Neuronal growth factors can help neurons survive, regenerate, and maintain themselves. NGF is a neurotrophin (N.T.s) gene family member, which encodes proteins with related structures and roles. A decrease in NGF synthesis affects the tiny fibers responsible for pain and temperature perception, which in turn influences the pathogenesis of diabetic neuropathy [8]. Clinical recommendations state that drugs such as amitriptyline and duloxetine, opioids, GABA derivatives like gabapentin and pregabalin, and topical medications like capsaicin can all be used to treat severe diabetic neuropathy. The FDA has approved tapentadol, duloxetine, and pregabalin to treat DPN [11]. Studies have revealed several adverse effects, including agitation and asterixis, with these traditional medications [12]. Additional adverse effects include constipation, nausea, vomiting, cardiac problems, hepatic malfunction, and somnolence [13]. Moreover, no total alleviation of neuropathic pain has been documented [14].

The aim of this study was to assess the safety and efficacy profile of the test drug *(e)-1-benzoyl3-((4-(dimethylamino) benzylidene) amino)-2-(4-(dimethylamino) phenyl)-2,3dihydroquinazoline-4(1h)-one* and to evaluate the influence of the test compound in nociception and diabetes-induced neuropathy and to assess the effect its effect on the liver tissues while examining its histopathological and biochemical parameters.

## Materials and methods

### Chemicals and equipment's

The main chemicals and equipment used in this research study were alloxan (Sigma, USA), Test compound (donated by Institute of Chemical Sciences, University of Peshawar), Gabapentin (Lowitt Pharma, Pakistan), Glibenclamide (Sanofi Aventis), Dimethyl Sulphoxide (BDH Chemicals, England), Von Frey filaments (Touch Test, USA), and Hotplate (Harvard apparatus).

### Ethical approval

Ethical committee of the Pharmacy Department of the University of Peshawar approved experimental methods (approval number 06/EC-FLES-UOP/2023). The testing period was from 8:00 am to 5:00 pm. The Scientific Procedure Act of U.K. 1986 and The ARRIVE Guidelines 2.0 were followed when doing procedures on animals [15].

### Animals handling

This study utilized albino mice (Balb-C strain), which were bred in the animal house of the University of Peshawar. The animals were provided with ad libitum access to water and a standardized diet. The animals were kept in well-ventilated housing maintaining temperature at 26 ° C and humidity between 30% and 70% and provided with 12:12 hour light-dark cycle to support the circadian rhythms of the animals. The housing was also supplied with dry wood shavings used as bedding, which were regularly changed to maintain a hygienic and microbe-free environment. They had continuous access to clean water and a standardized, nutritious diet to ensure their well-being. During the course of the experiment, animals were euthanized to collect tissue samples and to minimize any further distress or suffering. Euthanasia was

performed using the cervical dislocation method following the AVMA guidelines provided by American Veterinary Medical Association for humane euthanasia.

## Acute toxicity

Healthy experimental mice were selected for the acute toxicity study. The mice were administered the test compound intraperitoneally at doses of 10, 30, 50, 100, 200, and up to 300 mg/kg respectively. Toxicity assessments were conducted at 0, 30, and 60 minutes, followed by observations at 24 hours, 48 hours, 72 hours, and on the 7th day. The mice's were then observed for behavioral changes like writhing, aggression, etc. and mortality at regular intervals, i.e., 0, 30 min, 1 hour, 48 hours, and weekly [16]. None of the mice showed any of the above-mentioned signs. The 300 mg/kg dose was considered to be the safe dose as per research for acute toxicity conducted. The acute toxicity test was conducted on the 15th of November, 2023.

## Induction of diabetes mellitus

5% (w/v) alloxan solution was prepared by dissolving 5g in 100 ml normal saline in the absence of light. Diabetes was induced in experimental mice predistributed in groups by intraperitoneal injection of alloxan (150 mg/kg body weight) following a 16-hour food deprivation. Mice were allowed to resume eating 1 hour post-injection [17], developing full-blown diabetes within 72 hours [18]. The induction of diabetes using alloxan was done on the 22nd of November, 2023.

## Assessment of blood glucose level and body weight

Blood glucose levels of mice of all predetermined groups were monitored using a glucometer on days 0, 5, 15, and 29 after diabetes induction. Blood samples were taken using the tail-tip method. Body weight was measured for diabetic and saline-treated mice at the same intervals. Following diabetes induction, mice received glibenclamide (the standard drug) and the test compound [17].

## Pre-clinical models/tests for the pharmacological assessment of the test substance

**Anti-nociception & thermal test.** Mice weighing 18-22g and grouped (standard, disease control, tramadol 50 mg/kg, test 10 mg/kg & 20 mg/kg) were habituated to the laboratory for 1 hour. Pre-tested on a hot plate (54°C), mice with a response exceeding 15 seconds were excluded. After 30 minutes, mice received intraperitoneal injections of the test drug (10 & 20 mg/kg), tramadol (50 mg/kg), or saline. Latency to pain (hind limb lick, turn, or jump) was measured 30 and 60 minutes later. Ultimately, the percentage of analgesia was calculated using a formula [19]:

$$\text{Pecent Analgesic activity} = ((L_t - L_c)/L_c) \times 100$$

Where:
$L_t$ = latency time after treatment
$L_c$ = latency time before treatment (Control).

## Writhing test

For the writhing test, mice received an intraperitoneal injection of 1% acetic acid (10 ml/kg) to cause peripheral agony. The experiment was conducted on all five groups of mice. Food was withheld for two hours before being given an acetic acid injection. All animals were given specific medications, such as the standard medication (diclofenac 50 mg/kg), the test medication (10 and 20 mg/kg), and 0.9% normal saline followed by 1% acetic acid after 30 minutes. The frequency of body elongations and abdominal contractions [20] were then measured during a 20–30 minute interval following a 5-minute acetic acid treatment. Ultimately, a protection percentage was computed [19].

$$\% \text{ Protection} = \frac{1 - \text{mean no. of constrictions of drug incorporated group}}{\text{Mean number of constrictions in control}} \times 100$$

### Anti-neuropathy test/model

**Allodynia of static type.** Five predetermined groups of Albino/Balb-c mice were assessed for Static Allodynia. Using Von Frey filaments of varying forces, i.e., (0.4, 0.6, 1, 1.4, 2, 4, 6, 8, 10 and 15 g), static allodynia was evaluated on day 29. The mice were acclimatized 15–45 minutes before being examined for Static Allodynia. Von-Frey filaments were then applied perpendicularly to the plantar surface of the hind paw. The strain exerted was such that the filament would buckle for an interval of up to six seconds, or until the animal showed paw-licking activity or paw withdrawal, called paw withdrawal threshold (PWT). A 15-gram strength was deemed the cutoff point; no more force was used afterwards. Until there were four consecutive affirmative responses or five consecutive negative responses, the process was repeated. These strands were used a short while apart to avoid having the animal's performance affected by an earlier force [17]. The procedure was repeated after giving an I.P dose of the test compound (10, 20 mg/kg) and gabapentin (75 mg/kg) to their respective groups, and data was collected.

**Allodynia of dynamic type.** In five predetermined groups of mice, Dynamic Allodynia was assessed on the 29th day following the alloxan injection by lightly massaging the plantar surfaces of their hind paw using a cotton bud. Mice lift the hind paw for a particular time, termed paw withdrawal latency (PWL). The animals that exhibited a response within 8 seconds were chosen, with a 15-second time limit [18]. After giving the test compound (10, 20 mg/kg) and standard drug (gabapentin 75 mg/kg), the experiment was repeated to assess the dynamic allodynia for data comparison.

**Cold allodynia.** Cold Allodynia was evaluated in all five predetermined groups of mice. After acclimatizing the animal for 30 minutes, acetone (0.05 ml per spray) at 30-second intervals was sprayed thrice on the plantar surface of each hind paw through a syringe. The length of the paw withdrawal was measured [21]. Then, the test compound (10, 20 mg/kg) and gabapentin (75 mg/kg) were given I.P to their respective groups, and the experiment was conducted again for data comparison and analysis.

**Punctate hyperalgesia.** All five prearranged mouse groups were tested for punctate hyperalgesia. The mice were positioned on an elevated grid, and their hind paw plantar surfaces were rumpled with a standard safety pin until the paw skin buckled but did not puncture. Paw withdrawal time was then measured before and after giving test and standard drugs to their respective groups, with a time range of a minimum of 0.5 seconds to a maximum of 5 seconds [22].

**Heat hyperalgesia.** The experiment on Heat hyperalgesia was conducted for each of the five prearranged groups. The mice were put on a hot plate that had been heated to 52.0±0.2°C. The mice's latency of paw withdrawal (PWL) was recorded for 60 seconds by observing behaviours such as licking, flipping their hind limbs, and running away from the device [23]. The procedure was done before and after administering tests and standard drugs to their respective groups for data analysis.

### Histological and biochemical investigations

**Biochemical assessment.** For biochemical assessment, mice's blood samples were taken from each group and immediately transferred to EDTA tubes on ice for analysis. The samples were then centrifuged at 300 rpm to extract the serum, which was then stored at −8 °C until biological parameters like bilirubin, ALT, AST, and alkaline phosphatase were measured [24].

**Histological study.** Histological study were done on the experimented mice's liver. The liver was removed at the end of the experiment and preserved in a 10% formalin solution, which was studied under a microscope afterwards. The tissues were removed, filled with paraffin wax, and then cleaned with xylene before slicing into tiny five μm-thick slices. After staining the tissues with hematoxylin and eosin, they were mounted on slides so that the hepatocytes could be examined under a microscope [25].

## Statistical analysis

The statistical analysis was performed using the latest version IBM SPSS (Statistical Package for the Social Sciences) software. The necessary post-hoc analysis was conducted after the study's results were statistically assessed using ANOVA in the Graph Pad Prism program version 5.

## Results

### Impact of test compound and standard (Glibenclamide) in a diabetic model

Figure 1 **(Graphs A & B)** show that mice given 150 mg/kg of alloxan had significantly higher blood glucose levels than the saline group, indicating the beginning of diabetes mellitus. On day 29, promising results were seen after administering glibenclamide (at 5 mg/kg) and the test chemical (10 & 20 mg/kg). After 60 and 120 minutes, the test chemical at 20 mg/kg sfluignificantly reduced blood glucose levels in comparison to the glibenclamide group (standard). The data was gaged using one-way ANOVA followed by post-hoc analysis using the st-hocDunnett's test.

### Analgesic impact of standard (Tramadol) and test compound in hot plate test

Figure 1 (Graph C) compares the mice administered with the test chemical (10, 20 mg/kg) and standard (tramadol 50 mg/kg) at 30, 60, and 90 minutes to those given saline, a notable proportion of analgesia is seen when the hot plate test is performed on them. The analgesic effect on the mice is accentuated, with ***$P < 0.001$, **$P < 0.01$, and *$P < 0.05$ using one-way ANOVA followed by post-hoc analysis using the Dunnett's test.

### Analgesic impact of Diclofenac sodium (standard) and test compound in writhing test

Figure 1 **(Graph D)** depicts that after receiving a 1% acetic acid injection, mice exhibited a notable amount of abdominal writhing. Abdominal constrictions were significantly reduced in rats given conventional diclofenac sodium at 50 mg/kg and test compound at 10 & 20 mg/kg vs saline group. Each bar graph shows the statistical analysis and the number of writhes with ***$P < 0.001$, **$P < 0.01$, and *$P < 0.05$ assessed using one-way ANOVA followed by post-hoc analysis using the Dunnett's test.

### Results of the test compound in allodynia of static type

A more significant drop in the paw withdrawal threshold in albino mice treated with alloxan 150 mg/kg on day 5, 15, and 29th experimental days than in the group treated with saline suggested the onset of static allodynia is presented in Fig 1, representing the **Graphs E & F**. On day 29, promising results were noted with the administration of the test drug at 10, 20 mg/kg and standard gabapentin 75 mg/kg at 30, 60, and 90 minutes. Following a one-way ANOVA for data interpretation, the post hoc Dunnett's test was performed as specified.

### Results of the test compound in allodynia of dynamic type

Figure 1 **(Graphs G & H)** describes that the mice injected with 150 mg/kg of alloxan showed a marked reduction in paw withdrawal latency on days 5, 15 and 29 in comparison with the normal saline group, suggesting the beginning of dynamic allodynia. On the 29th day, after giving the test drug at 10, 20 mg/kg and standard gabapentin at 75 mg/kg, favourable results were seen. The likelihood values that result from a one-way ANOVA and Dunnett's test are *$P < 0.05$, **$P < 0.01$, and ***$P < 0.001$ are shown in **graph H.**

### Impact of test drug in cold allodynia

Figure 1 **(Graph I & J)** shows the comparison of mice treated with saline, which exhibit the onset of cold allodynia; animals given 150 mg/kg of alloxan exhibited a markedly longer duration of paw withdrawal. In contrast to the alloxan-induced

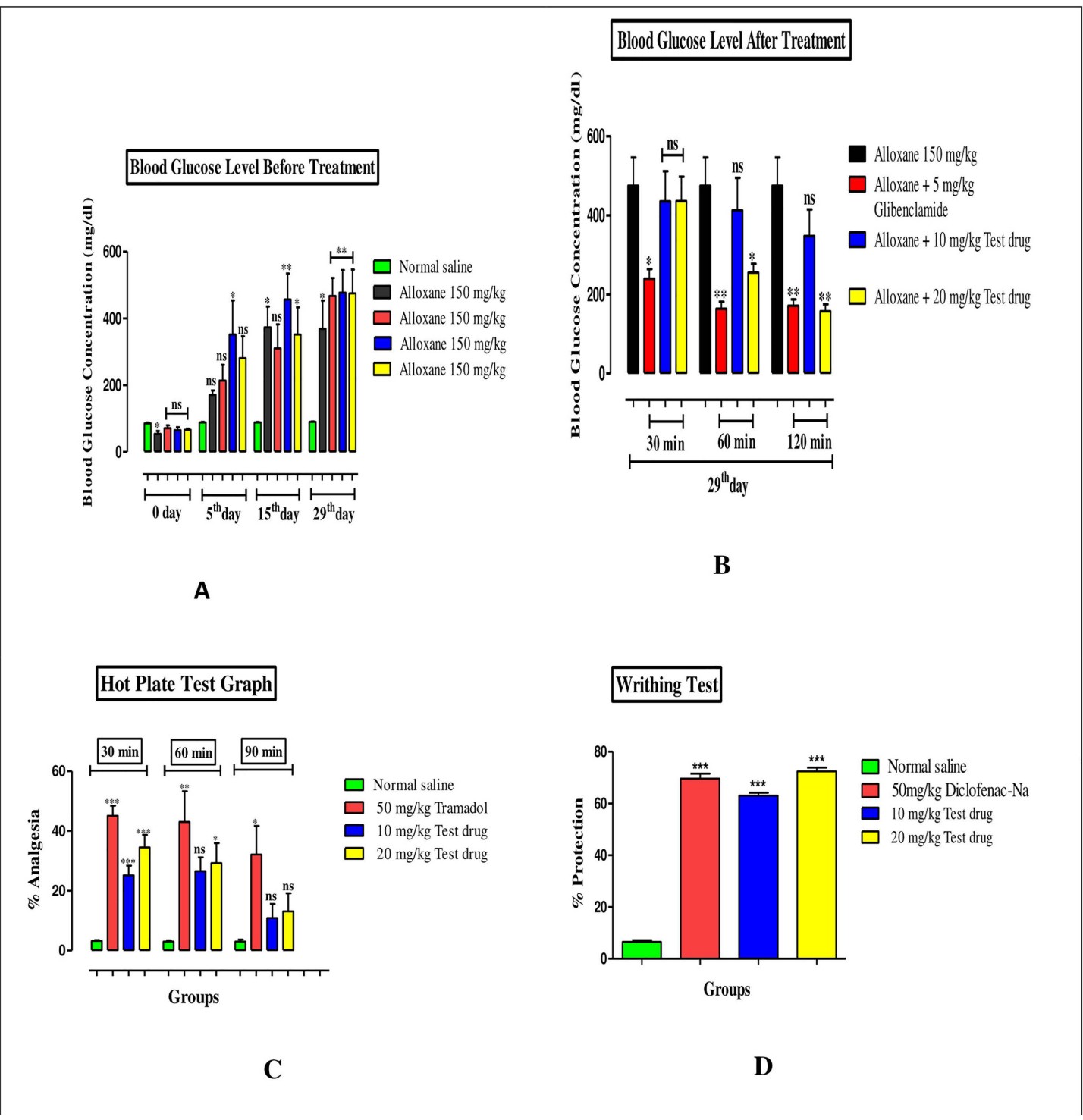

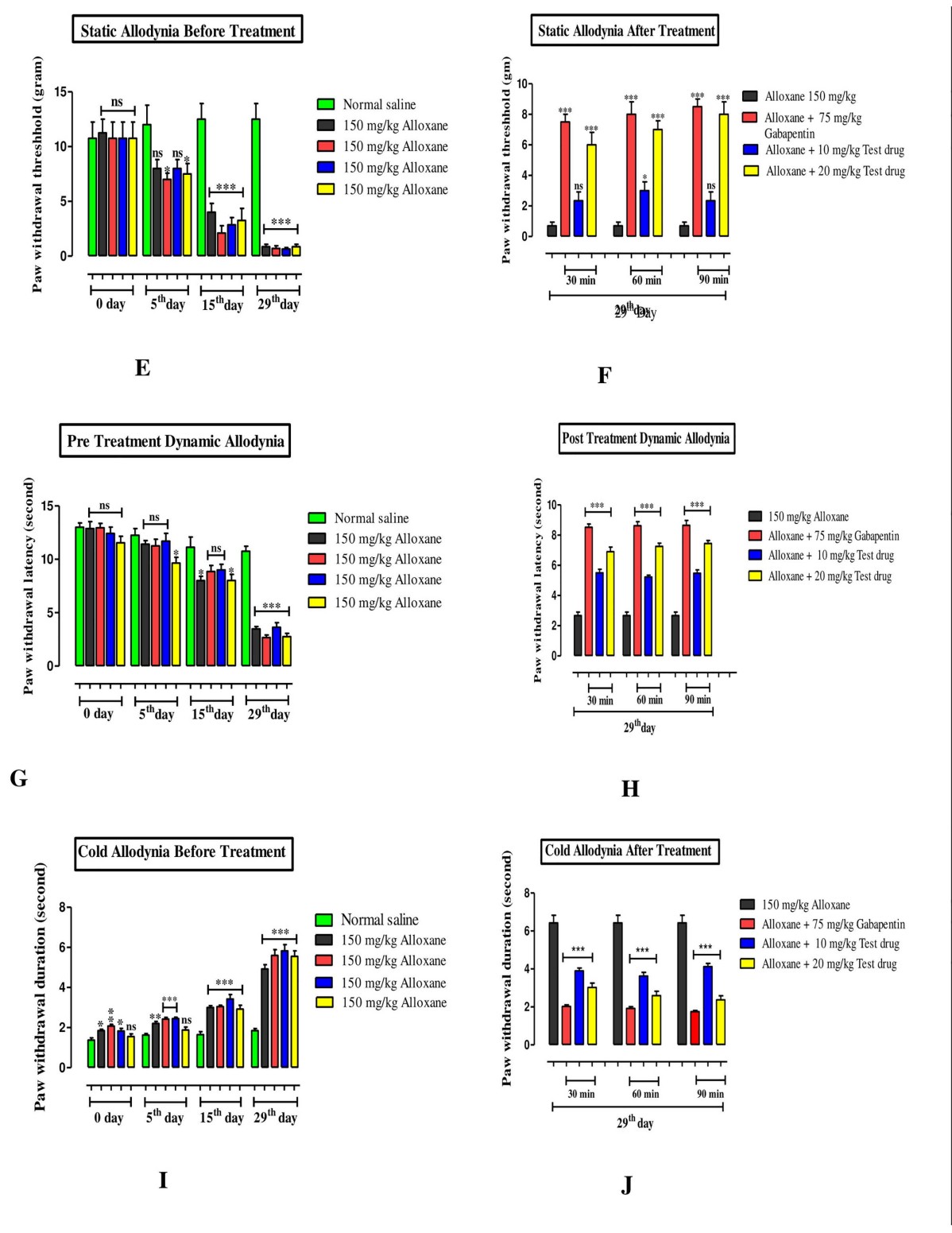

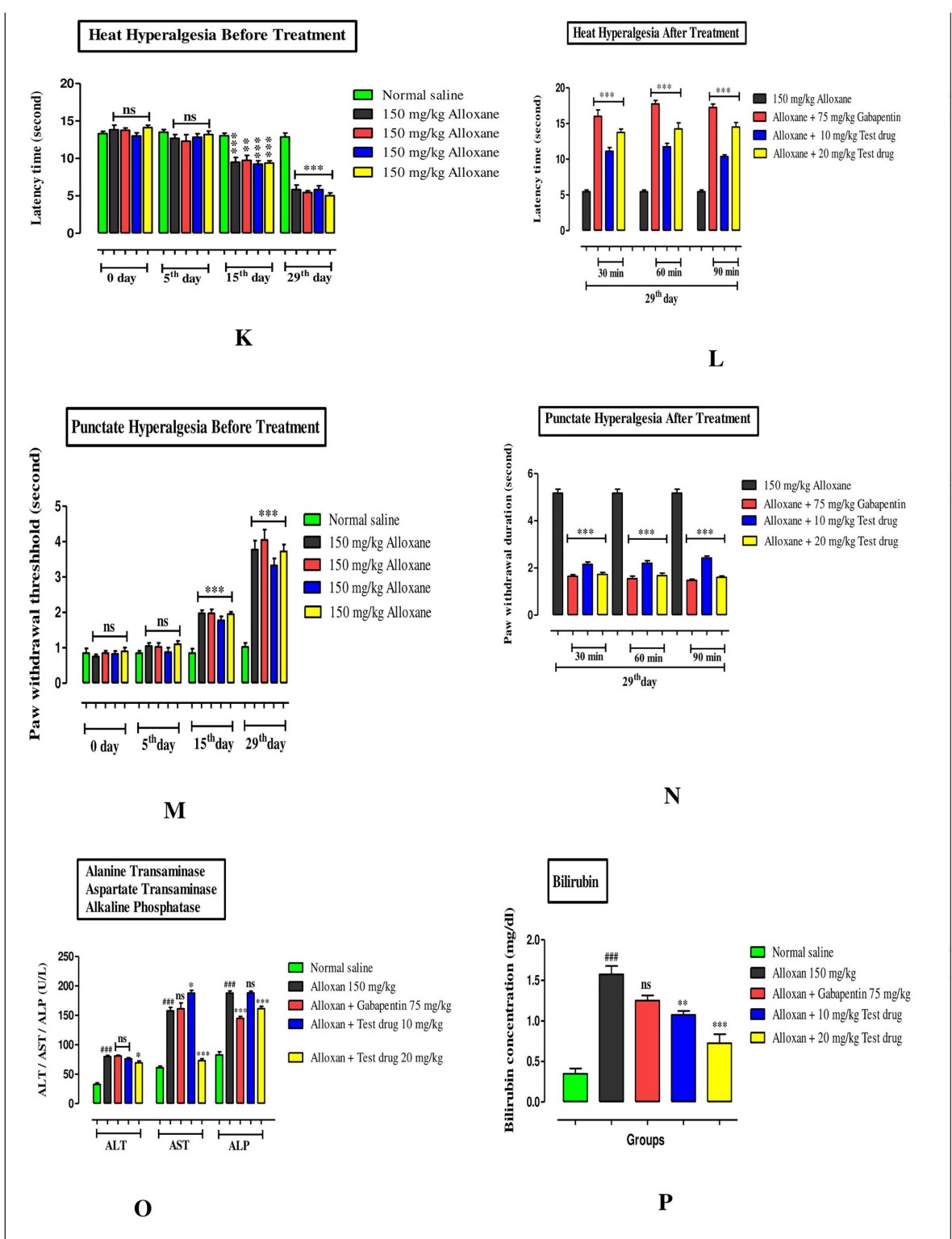

**Fig 1. (Graph A to P) Represent experimental procedures/protocols followed.**

rise, the PWD was significantly decreased on day 29 while administering the test drug at 10, 20 mg/kg and standard gabapentin at 75 mg/kg. The comparison of after treatment was shown in **graph j** as compared to **graph I**, denoted by P<0.001(***), assessed using one-way ANOVA followed by post-hoc analysis using the Dunnett's test.

### Results of test compound in heat hyperalgesia model

When animals were injected with alloxan, their latency times significantly decreased, particularly on days 15 and 29, indicating the development of heat hyperalgesia in animals. After treatment with test compound (10, 20 mg/kg) and 75 mg/kg of gabapentin (standard) resulted in a considerable upsurge in latency time.

In Fig 1**, Graph—K** demonstrates that in contrast to a saline control group, mice on days 0, 5, 15, and 29 prior to treatment display heat hyperalgesia. **Graph—L** illustrates the effects of the test compound at (10, 20 mg/kg) and standard (gabapentin 75 mg/kg) at 30, 60, and 90 minutes. A significant level of ***P<0.001 was found using one-way ANOVA and Dunnett's analysis when compared to the group that received alloxan treatment.

### Effects of test compound in punctate hyperalgesia

Figure 1 **(Graphs M and N)** depicts about those experimental animals to which 150 mg/kg injections of alloxan were administered; their paw withdrawal threshold significantly decreased compared to the group receiving saline treatment, indicating the development of punctate hyperalgesia. Effective outcomes were obtained by administering the test drug at 10, 20 mg/kg & gabapentin 75 mg/kg on the 29th day at 30, 60, and 9 minutes as PWT increased significantly. A one-way ANOVA and a post hoc Dunnett's test yielded a significance level of ***P<0.001.

### Biochemical study of liver

On the 29th experimental day, many biochemical parameters, including aspartate transaminase, bilirubin, alanine transaminase, and alkaline phosphatase, were measured in blood samples from predetermined mouse groups. The liver profile or biochemical parameters were done to assess the hepatic concerns related to the use of each type of drug. Both tabular and graphical representations of the biochemical data are provided in Table 1 and Fig 1 **(Graphs O &P)**.

### Graph study of the standard and test compound on liver biochemical parameters

Fig 1, highlighting graphs O and P, demonstrates the mice that received 150 mg/kg of alloxan revealed a substantial elevation in liver biochemicals (ALP, ALT and AST) in comparison to a normal saline group, suggesting that alloxan has a deleterious effect on liver function. The test drug, administered at a dosage of 20 mg/kg, showed a significant reduction in cases of AST and ALP and a slight decrease in ALT levels when compared to animals treated with alloxan. On the other hand, gabapentin had a notable impact in ALP instances.

A one-way ANOVA and post hoc test were used to examine the data; *P<0.05, ***P<0.001 showed the statistical significance as shown in **(Graph—O)**. The blood levels of bilirubin were higher in the alloxan-treated animals than in the saline group. Analysis of variance and post hoc Dunnett's test demonstrated a significant reduction in bilirubin concentration at 10 and 20 mg/kg of the test drug; the statistical impact is ***P<0.001, **P<0.01 represented in **(Graph—P)**.

### Histological study

### Changes in liver histology treated with normal saline, standard drug and test compound

The liver histological appearance was normal in the group that received normal saline administration. There was a central vein (CV) encircled by intact endothelium in the liver lobule. Hepatic plates with mono- and multi-nuclear hepatocytes, divided by transparent sinusoids, are typically visible around the CV. Glycogen granules scattered throughout the cytoplasm of the hepatocytes gave the impression of being filled Fig 2 (A). The hepatic lobule of the diabetes control group

**Table 1. Comparison of biochemical indicators followed (Normal Saline NS, Disease Control DC, Standard Drug SD, and Test Drug T.D.) ingestion.**

| Sr. No | Animals | AST (U/L) | ALT (U/L) | ALP (U/L) | Bilirubin (mg/dl) |
|---|---|---|---|---|---|
| 1 | NS1 | 40 | 34 | 97 | 0.4 |
| 2 | NS2 | 43 | 31 | 84 | 0.2 |
| 3 | NS3 | 35 | 27 | 72 | 0.3 |
| 4 | NS4 | 37 | 32 | 78 | 0.5 |
| 5 | DC1 | 154 | 76 | 190 | 1.6 |
| 6 | DC2 | 144 | 81 | 189 | 1.3 |
| 7 | DC3 | 161 | 85 | 177 | 1.8 |
| 8 | DC4 | 172 | 78 | 195 | 1.6 |
| 9 | SD1 | 159 | 84 | 147 | 1.4 |
| 10 | SD2 | 135 | 80 | 138 | 1.1 |
| 11 | SD3 | 170 | 82 | 153 | 1.2 |
| 12 | SD4 | 181 | 78 | 142 | 1.3 |
| 13 | TD1 | 193 | 80 | 194 | 1 |
| 14 | TD2 | 185 | 74 | 187 | 1 |
| 15 | TD3 | 176 | 78 | 180 | 1.2 |
| 16 | TD4 | 197 | 72 | 191 | 1.1 |
| 17 | TD5 | 81 | 77 | 156 | 0.5 |
| 18 | TD6 | 74 | 69 | 166 | 0.8 |
| 19 | TD7 | 72 | 61 | 154 | 1 |
| 20 | TD8 | 65 | 70 | 170 | 0.6 |

exhibited more severe histological alterations. Severe central venous congestion containing necrotic debris and red blood cells was noted.

Additionally, the endothelium lining the central vein seemed to be detached and injured. Additionally, there were dilated sinusoidal gaps with lymphocytes penetrating them. Hepatocyte necrosis occurred sporadically. The majority of the hepatic lobules had microvesicular steatosis Fig 2 (B) (Alloxan administered group).

Significant liver histological alterations were also observed in the standard drug group. The central vein displayed endothelial exfoliation and congestion with necrotic debris. Sinusoidal dilatation and congestion were noted. Moreover, there was microvesicular steatosis. Hepatocytes showed minor necrosis, and several hepatic lobules looked damaged Fig 2 (C). Hepatocytes in the test group (10 mg/kg) had mild to moderate cytoplasmic degranulation and otherwise seemed almost normal. Red blood cells seemed to be clogging the central vein, and a slight endothelium separation was visible. Although the sinusoids also appeared normal, a few red blood cells could be seen in their lumen Fig 2 (D). Although there were some minor histological alterations, the test group's (20 mg/kg) hepatic lobule likewise had a normal histological appearance. Despite having a healthy endothelium, the central vein was clogged with red blood cells. The undamaged hepatocytes had modest glycogen degranulation, and the sinusoidal gaps were regularly spaced Fig 2 (E).

## Discussion

The test chemical *(e)-1-benzoyl-3-((4- (dimethylamino) benzylidene) amino)-2-(4-(dimethylamino) phenyl)-2, 3dihydroquinazoline-4(1h)-one* anti-diabetic effect was compared to that of standard glibenclamide in an alloxan-induced diabetes model. When given at a dose of 20 mg/kg, the test chemical dramatically lowered the blood glucose level in diabetic mice. The test substances at 10 & 20 mg/kg were checked against the usual doses of diclofenac sodium & tramadol (50 mg/kg each) to assess the analgesic effects on the central and peripheral nervous systems using methods such as the

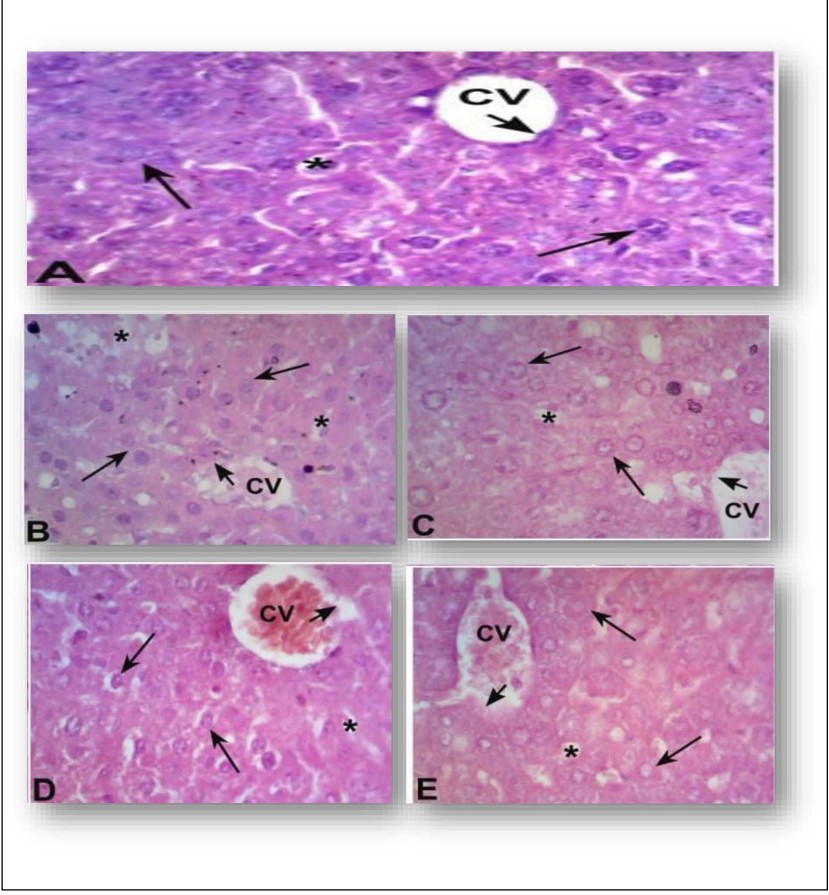

**Fig 2. (Graph 2A to 2E) Normal Histological Appearance of Mice Liver: (A)**.The hepatic lobule's histological aberrations were worse in the diabetic control group **(B)**. The standard group displayed notable histological alterations in the liver **(C)**. Test group (10 mg/kg) hepatocytes with mild to severe cytoplasmic degranulation **(D)** seemed nearly normal. The test group (20 mg/kg) had modest histological alterations but otherwise showed normal histological appearance **(E).**

hot plate and writhing test. Regarding writhing in animals, the test substance at 10 & 20 mg/kg showed a significant (%) protection compared to the writhing of animals in the saline group. In the hot plate test, the test compound at both doses proved significant (%) analgesia.

Similarly, the Alloxan-induced diabetic pain models, such as static and dynamic allodynia, heat hyperalgesia, cold allodynia, and punctate hyperalgesia, were used to study a drug's anti-neuropathic effects. Comparing the chosen test chemical to regular gabapentin, whose usage is restricted because of complex side effects such as agitation, asterixis, hepatic malfunction, sexual dysfunction, and partial relief from neuropathic pain, it demonstrated a significant anti-neuropathic impact. The test chemical in this investigation significantly raised the paw withdrawal latency in both dynamic allodynia and heat hyperalgesia and the paw withdrawal threshold in both static and punctate allodynia at doses of 10 and 20 mg/ kg. The test chemical considerably lengthened the duration of paw withdrawal in cases of cold allodynia compared to the gabapentin and disease control groups.

Upon biochemical examination, the test substance at a dose of 20 mg/kg considerably reduced the concentrations of AST and ALP while only marginally lowering the ALT level. Only the standard gabapentin (75 mg/kg I.P) considerably reduced ALP (Table 1). As indicated in Table 1, the test medication at both doses significantly lowers the bilirubin

concentration compared to the alloxan-induced model. Moreover the test drug did not exhibit any notable adverse effects cmapred to the standard (gabapentin 75 mg/kg I.P). The gabapentin group exhibited notable liver histological alterations, according to the histopathology investigation. The central vein displayed endothelial exfoliation and congestion with necrotic debris. Sinusoidal dilatation and congestion were noted. Moreover, there was microvesicular steatosis. Fatty microvesicles smaller than 1 µm were found in the hepatocytic cytoplasm in cases of microvesicular steatosis [25,26]. Hepatocytes showed signs of moderate necrosis, and some hepatic lobules looks damaged also. The test substance appeared almost normal and caused mild to severe cytoplasmic degranulation in hepatocytes when administered at a dose of 10 mg/kg.

## Conclusion

The test compound (e)-1-benzoyl-3-((4- (dimethylamino) benzylidene) amino)-2-(4-(dimethylamino) phenyl)-2,3 dihydroquinazoline-4(1h)-one was proven to be both safe and effective against diabetes-induced neuropathy assessed through preclinical pain model. Comparing it to the common medications Glibenclamide, Diclofenac sodium, Tramadol, and Gabapentin at their recommended doses, it shows favorable anti-diabetic, anti-nociceptive, and anti-neuropathic effects. Additionally, it was determined that the test drug did not exhibit any notable adverse effects. The chosen compound is also a ketone by nature, and ketone bodies and their derivatives exhibit a variety of pharmacological properties, including the ability to mitigate neurodegenerative symptoms. Compared to the standard drug gabapentin, which showed obvious changes in hepatic histopathology, the test compound at 10 and 20 mg/kg displayed normal hepatocyte appearance with only mild to moderate histological changes.

## Author contributions

**Conceptualization:** Haider Ali, Asif Jan, Syed Shaukat Ali.

**Data curation:** Haider Ali, Gowhar Ali, Abdur Rahim.

**Formal analysis:** Haider Ali, Asif Jan, Gowhar Ali, Syed Shaukat Ali, Abdur Rahim.

**Investigation:** Asif Jan, Gowhar Ali, Abdur Rahim, Ursula Abu Nahla.

**Methodology:** Haider Ali, Asif Jan, Abdur Rahim, Ursula Abu Nahla.

**Project administration:** Ursula Abu Nahla.

**Resources:** Ursula Abu Nahla.

**Supervision:** Gowhar Ali, Ursula Abu Nahla.

**Validation:** Haider Ali, Gowhar Ali, Syed Shaukat Ali.

**Visualization:** Syed Shaukat Ali.

**Writing – original draft:** Asif Jan, Syed Shaukat Ali.

**Writing – review & editing:** Asif Jan, Gowhar Ali.

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
