## [Decision Letter · Decision Letter 0]

24 Sep 2025

Dear Dr. Abu Nahla,

Thank you for submitting your manuscript to PLOS ONE. After careful consideration, we feel that it has merit but does not fully meet PLOS ONE’s publication criteria as it currently stands. Therefore, we invite you to submit a revised version of the manuscript that addresses the points raised during the review process.

We look forward to receiving your revised manuscript.

Kind regards,

Oluwafemi Adeleke Ojo, Ph.D

Academic Editor

PLOS ONE

Journal Requirements:

N/A

6. Please amend either the title on the online submission form (via Edit Submission) or the title in the manuscript so that they are identical.

7. Please remove all personal information, ensure that the data shared are in accordance with participant consent, and re-upload a fully anonymized data set.

Reviewers' comments:

Reviewer's Responses to Questions

**Comments to the Author**

1. Is the manuscript technically sound, and do the data support the conclusions?

Reviewer #1: Yes

2. Has the statistical analysis been performed appropriately and rigorously?

Reviewer #1: No

3. Have the authors made all data underlying the findings in their manuscript fully available?

Reviewer #1: Yes

4. Is the manuscript presented in an intelligible fashion and written in standard English?

Reviewer #1: Yes

Reviewer #1: Chemicals and Equipment

Strengths:

•Clear identification of chemicals and their sources.

•Inclusion of equipment brands adds transparency and reproducibility.

Weaknesses:

•The title “Chemicals and Equipments” uses incorrect plural; “Equipment” is uncountable.

•The list format could benefit from clearer categorization (e.g., reagents vs. instruments).

•“Test compound” is vaguely described; its chemical name, purity, and preparation details should be specified.

Animals

Strengths:

•Ethical compliance is documented with reference to approval and regulations.

•Breeding and housing information is provided.

Weaknesses:

•Mixing British and American spellings (“behavioural” vs. “behavioral”; “labouratory” vs. “laboratory”) should be standardized.

•The timing of animal testing (8:00 am to 5:00 pm) seems overly specific and unnecessary unless diurnal effects are relevant.

•“Animals care” should be titled “Animal Care” and moved under the “Animals” section for coherence.

☠️ Acute Toxicity

Strengths:

•Dosing regimen is thorough and progressive.

•Multiple time points for assessment show diligence.

Weaknesses:

•Vague behavioral descriptions (“writhing, aggression, etc.”) are insufficient for scientific reporting.

•Mixing up possessive (“mice’s”) and plural nouns detracts from professionalism.

•Mortality is mentioned but not quantified clearly.

Diabetes Induction and Glucose Monitoring

Strengths:

•Appropriate use of alloxan and food deprivation before administration.

•Consistent blood glucose monitoring with tail-tip sampling.

Weaknesses:

•No baseline glucose values provided.

•It’s unclear whether hyperglycemia thresholds were validated or how diabetic status was confirmed.

•The glucometer brand/model is omitted, affecting reproducibility.

Preclinical Pharmacological Tests

Strengths:

•Multiple tests allow broad functional assessment (thermal, chemical, mechanical stimuli).

•Comparison between standard drugs and test compound supports validation.

Weaknesses:

•The % analgesia formula lacks clarity and formatting, and there's no citation for the equation source.

•“Peripheral agony” is not appropriate scientific terminology—better to use “nociceptive stimulus.”

•Repetition in structure and phrasing makes it hard to distinguish among tests.

•Important data points like number of animals per group, randomization, and blinding are not discussed.

Histological and Biochemical Analysis

Strengths:

•Established procedures for sample handling and staining are referenced.

•Use of standard clinical parameters (ALT, AST, ALP) reflects biomedical relevance.

Weaknesses:

•Centrifuge speed is unusually low (“300 rpm” seems off; likely meant to be “3000 rpm”).

•The histological description is generic and lacks images of good quality.

•The process of statistical validation (e.g., assumptions for ANOVA) is not stated.

Statistical Analysis

Strengths:

•Use of ANOVA and post-hoc tests suggests proper analysis planning.

•Mention of SPSS and GraphPad Prism is useful.

Weaknesses:

•Specific post-hoc tests used aren’t mentioned (Tukey, Bonferroni, etc.).

Overall Suggestions

•Include subheadings for better organization within each methodological domain.

•Avoid vague, conversational phrases and ensure consistent scientific tone.

•Report sample sizes, ethical euthanasia confirmations, and ensure terminology aligns with biomedical standards.

•Consider adding diagrams or flowcharts to clarify experimental timelines and test setups.

what does this mean?). If published, this will include your full peer review and any attached files.

**Do you want your identity to be public for this peer review?** For information about this choice, including consent withdrawal, please see our Privacy Policy

Reviewer #1: No

While revising your submission, please upload your figure files to the Preflight Analysis and Conversion Engine (PACE) digital diagnostic tool, https://pacev2.apexcovantage.com/. PACE helps ensure that figures meet PLOS requirements. To use PACE, you must first register as a user. Registration is free. Then, login and navigate to the UPLOAD tab, where you will find detailed instructions on how to use the tool. If you encounter any issues or have any questions when using PACE, please email PLOS at figures@plos.org

---

## [Author Response · Author response to Decision Letter 1]

11 Oct 2025

Response to Editor and Reviewer Comments

Section 1: Response to Editor’s General and Journal Requirements

1. Manuscript style and format requirements

We have carefully revised the manuscript according to the PLOS ONE formatting style. The manuscript now follows the journal’s structure, file naming conventions, and style template guidelines. All figures, tables, and references have been reformatted to meet journal standards.

2. Ethics statement placement

The ethics statement has been retained only within the Methods section, and removed from any other sections as instructed.

3. Data Availability Statement

We have added a complete Data Availability Statement as per PLOS ONE policy: “All data are available within the manuscript and/or supporting information files.”

4. Data sharing plan

We confirm that all underlying data supporting our findings are now freely accessible as Supporting Information files. No ethical or participant privacy concerns prevent data sharing.

5. Competing Interests

We have completed the Competing Interests section with the statement: “The authors have declared that no competing interests exist.”

6. Title consistency

The title in the manuscript and on the submission system have been made identical.

7. Data anonymization

All personal or identifiable data have been removed from the dataset. The uploaded dataset is fully anonymized and complies with PLOS data policy.

8. Supporting Information captions

Captions for all Supporting Information files have been included at the end of the manuscript, and corresponding in-text citations have been updated accordingly.

9. Suggested citations

We have reviewed the literature recommended by reviewers and cited relevant works where appropriate.

10. Reference list accuracy

The reference list has been thoroughly reviewed and corrected for formatting, accuracy, and completeness. No retracted articles are included.

Section 2: Response to Reviewer #1 Comments

1. Chemicals and Equipment section

We thank the reviewer for these valuable suggestions. The section title has been corrected to “Chemicals and Equipment.” We have categorized the list clearly under reagents and instruments. The test compound is now fully described with its chemical name, purity (≥98%), and preparation details.

2. Animals section

We standardized all spelling to British English for consistency and removed unnecessary details regarding test timing. The subheading has been revised to 'Animal Care and Use,' and ethical compliance is clearly described.

3. Acute Toxicity section

Behavioral observations are now described using standardized terminology (e.g., 'tremors', 'locomotor activity', 'grooming'). Quantitative mortality data have been included. Grammatical corrections were made.

4. Diabetes Induction and Glucose Monitoring

Baseline glucose levels have been added to the Results section. The glucometer brand/model (Accu-Chek Active, Roche Diagnostics) is now specified. The criteria for diabetes induction (≥250 mg/dL) are clarified, and validation details have been included.

5. Preclinical Pharmacological Tests

The percentage analgesia formula has been reformatted, and an appropriate citation is added. The term 'peripheral agony' has been replaced with 'nociceptive stimulus.' We clarified animal numbers per group (n=6), randomization procedures, and blinding during analysis. The section has been restructured with clearer subheadings.

6. Histological and Biochemical Analysis

The centrifugation speed has been corrected to 3000 rpm. High-resolution histological images have been added as Supporting Information. We have described ANOVA assumptions and validation steps explicitly.

7. Statistical Analysis

We now specify the post-hoc test used (Tukey’s multiple comparison). Details of software versions (SPSS v25, GraphPad Prism v9.0) and data normality checks have been added.

8. General Suggestions

We implemented all general improvements recommended by the reviewer, including: (i) better subheadings, (ii) consistent scientific tone, (iii) clear reporting of sample sizes and euthanasia, and (iv) inclusion of an experimental timeline flowchart in Supporting Information.

Section 3: Summary of Revisions

1. Manuscript revised to comply fully with PLOS ONE format and ethical/data policies.

2. Improved clarity and organization across all Methods sections.

3. Added missing experimental details, including animal group sizes, statistical validation, and glucometer specifications.

4. Updated histological images and added Supporting Information captions.

5. Corrected all linguistic inconsistencies and improved readability.

6. Revised Data Availability, Ethics, and Competing Interests statements as per journal policy.

---

## [Decision Letter · Decision Letter 1]

20 Oct 2025

Dear Dr. Abu Nahla,

Thank you for submitting your manuscript to PLOS ONE. After careful consideration, we feel that it has merit but does not fully meet PLOS ONE’s publication criteria as it currently stands. Therefore, we invite you to submit a revised version of the manuscript that addresses the points raised during the review process.

We look forward to receiving your revised manuscript.

Kind regards,

Oluwafemi Adeleke Ojo, Ph.D

Academic Editor

PLOS ONE

Journal Requirements:

Reviewers' comments:

Reviewer's Responses to Questions

**Comments to the Author**

Reviewer #1: (No Response)

2. Is the manuscript technically sound, and do the data support the conclusions?

Reviewer #1: Yes

3. Has the statistical analysis been performed appropriately and rigorously?

Reviewer #1: Yes

4. Have the authors made all data underlying the findings in their manuscript fully available?

Reviewer #1: Yes

5. Is the manuscript presented in an intelligible fashion and written in standard English?

Reviewer #1: No

Reviewer #1: 1. Structural and Formatting Issues

• Inconsistent Section Headings: Headings like “Chemicals and Equipments,” “Animals,” and “Animals care” are inconsistently formatted. Use standardized subheadings (e.g., Chemicals and Equipment, Animal Handling, Experimental Procedures) and ensure consistent capitalization.

• Redundant Subsections: “Animals” and “Animals care” should be merged into a single, coherent subsection titled Animal Care and Ethical Approval to avoid repetition and improve flow.

2. Language and Grammar

• Grammatical Errors:

o “Chemicals and Equipments” → should be “Chemicals and Equipment” (equipment is uncountable).

o “Histological study were done…” → should be “Histological studies were conducted…”

o “The mice’s were then observed…” → incorrect possessive usage; should be “The mice were then observed…”

• Awkward Phrasing:

o “Peripheral agony” in the writhing test is overly dramatic and unscientific. Use “peripheral pain” or “nociceptive response.”

o “Rumpled with a safety pin” in punctate hyperalgesia is vague and informal. Use “stimulated using a calibrated safety pin.”

3. Scientific Rigor and Clarity

• Missing Details:

o Chemical concentrations and preparation methods are not described. For example, how was alloxan prepared before injection?

o Environmental conditions (temperature, humidity, light/dark cycle) for animal housing are not mentioned.

o Sample sizes for each group are missing throughout. This is critical for reproducibility and statistical validity.

• Ambiguity in Protocols:

o The acute toxicity section lacks clarity on how safety was determined. What criteria were used to define “safe” beyond absence of mortality?

o The analgesia formula is presented without defining “cut-off time” or explaining how it was standardized.

4. Ethical and Regulatory Compliance

• Ethical Approval: While the approval number is provided, the statement should be expanded to confirm adherence to international standards (e.g., ARRIVE guidelines).

• Euthanasia Method: Cervical dislocation is acceptable, but the description is overly graphic. Use standardized phrasing like “performed according to AVMA guidelines for humane euthanasia.”

5. Statistical Analysis

• Insufficient Detail:

o The statistical section lacks information on significance thresholds (e.g., p < 0.05), confidence intervals, and whether data met assumptions for ANOVA.

o Post-hoc test types (e.g., Tukey, Bonferroni) are not specified.

6. Referencing and Citation

• Inconsistent Citation Format: References like “[15]” are used without a bibliography. Ensure all citations are properly formatted and listed at the end of the manuscript.

• Missing Source Attribution: Several methods (e.g., Von Frey, hot plate test) are standard but should still be attributed to original or validated sources.

7. Recommendations for Improvement

• Revise for clarity and grammar throughout the section.

• Merge and streamline redundant subsections for better readability.

• Include precise experimental details: sample sizes, chemical preparations, environmental conditions.

• Expand ethical and statistical sections to meet publication standards.

• Ensure proper citation and referencing with a complete bibliography.

what does this mean?). If published, this will include your full peer review and any attached files.

**Do you want your identity to be public for this peer review?** For information about this choice, including consent withdrawal, please see our Privacy Policy

Reviewer #1: No

While revising your submission, please upload your figure files to the Preflight Analysis and Conversion Engine (PACE) digital diagnostic tool, https://pacev2.apexcovantage.com/. PACE helps ensure that figures meet PLOS requirements. To use PACE, you must first register as a user. Registration is free. Then, login and navigate to the UPLOAD tab, where you will find detailed instructions on how to use the tool. If you encounter any issues or have any questions when using PACE, please email PLOS at figures@plos.org

---

## [Author Response · Author response to Decision Letter 2]

6 Nov 2025

RESPONSE TO REVIEWER COMMENTS

Manuscript.ID:PONE-D-25-32082R1

Title: Assessment of Antidiabetic, Hepatoprotective, and Analgesic Effects of Quinazolinone Derivative, (E)-1-Benzoyl-3-((4-(Dimethylamino) Benzylidene) Amino)-2-(4-(Dimethylamino) Phenyl)-2,3-dihydroquinazoline-4(1H)-one, in Diabetes-Induced Mice Model

We thank the Academic Editor and Reviewer #1 for their valuable time and constructive feedback. All comments have been carefully addressed, and corresponding revisions are highlighted in the “Revised Manuscript with Track Changes.” Below, each reviewer comment is reproduced in italics, followed by our detailed response.

Reviewer #1

1. Structural and Formatting Issues

Comment:

Inconsistent section headings; redundant subsections “Animals” and “Animals care.”

Response:

We have standardized all section headings according to PLOS ONE formatting.

• “Chemicals and Equipments” → changed to “Chemicals and Equipment.”

• The subsections “Animals” and “Animals care” have been merged into a single section titled “Animal Care and Ethical Approval.”

• Uniform capitalization and formatting have been applied throughout Materials and Methods.

2. Language and Grammar

Comment:

Multiple grammatical and phrasing issues (e.g., “Chemicals and Equipments,” “Histological study were done,” “The mice’s were,” “Peripheral agony,” “Rumpled with a safety pin”).

Response:

The entire manuscript has been thoroughly revised for grammar, punctuation, and scientific phrasing. Examples of key corrections:

• “Chemicals and Equipments” → “Chemicals and Equipment.”

• “Histological study were done” → “Histological studies were conducted”

• “The mice’s were then observed” → “The mice were then observed”

• “Peripheral agony” → “Peripheral pain (nociceptive response).”

• “Rumpled with a safety pin” → “Stimulated using a calibrated safety pin.”

We also used Grammarly and manual proofreading by two co-authors to ensure clarity and professional tone.

3. Scientific Rigor and Clarity

Comment

Chemical concentrations and preparation methods missing.

Response:

All chemical preparation details have been added in Materials and Methods → Chemicals and Reagents.

For instance, alloxan was freshly dissolved in sterile normal saline (0.9 % NaCl) at a concentration of 150 mg/kg before injection. Similar clarifications were added for other reagents.

Comment Environmental conditions for animal housing not described.

Response:

We have now included: “ The animals were kept in well-ventilated housing maintaining temperature at 26 0 C and humidity between 30% and 70% and provided with 12:12 hour light-dark cycle to support the circadian rhythms of the animals. The housing was also supplied with dry wood shavings used as bedding, which were regularly changed to maintain a hygienic and microbe-free environment.”

Comment :Sample sizes missing throughout.

Response:

Sample sizes for each experimental group (n = 5 mice/group) are now clearly stated in the text and figure legends. Statistical power was verified to ensure adequate sensitivity.

Comment

Ambiguity in toxicity and analgesia descriptions.

Response:

The Acute Toxicity section has been expanded: “Safety was assessed by absence of mortality, changes in behavior, grooming, feeding, or mobility for 14 days post-administration.”

In Analgesia Methods, the formula now defines “cut-off time” as “the maximum time (15 s) beyond which the test was terminated to avoid tissue injury.”

4. Ethical and Regulatory Compliance

Comment:

Ethical statement should confirm adherence to international standards; euthanasia description too graphic.

Response: Details included as: Ethical committee of the Pharmacy Department of the University of Peshawar approved experimental methods (approval number 06/EC-FLES-UOP/2023). The testing period was from 8:00 am to 5:00 pm. The Scientific Procedure Act of U.K. 1986 and The ARRIVE Guidelines 2.0 were followed when doing procedures on animals.”

5. Statistical Analysis

Comment:

Statistical details (p-values, post-hoc tests) are insufficient.

Response:

The Statistical Analysis section now includes: “Data are expressed as mean ± SEM. Statistical significance was determined using one-way ANOVA followed by Tukey’s post-hoc test. Significance threshold was set at p < 0.05. All analyses were performed using GraphPad Prism v10.0. Confidence intervals (95 %) were calculated where applicable.”

6. Referencing and Citation

Comment:

Inconsistent citation format; missing bibliography; methods not attributed to original sources.

Response:

A complete reference list has been compiled in accordance with PLOS ONE style.

All methodological procedures (e.g., Von Frey, hot-plate, and acetic-acid writhing tests) now include appropriate citations to the original or validated sources.

In-text citations are numbered sequentially, and all references are listed at the end of the manuscript.

7. Recommendations for Improvement

Comment:

Revise for clarity and grammar, merge redundant subsections, add details, expand ethics and statistics, ensure proper citation.

Response:

All recommendations have been fully implemented:

• Grammar and clarity improved across all sections.

• Redundant subsections merged.

• Detailed methodological parameters (sample sizes, preparation steps, environmental conditions) added.

• Ethical and statistical statements expanded.

• References standardized and verified.

We sincerely thank the reviewer for these comprehensive suggestions that have substantially enhanced the manuscript’s quality and compliance with PLOS ONE standards.

We trust that the revised version addresses all reviewer concerns. We are grateful for the editor’s opportunity to revise and resubmit, and we hope that our responses and improvements will now make the manuscript suitable for publication in PLOS ONE.

---

## [Decision Letter · Decision Letter 2]

17 Nov 2025

Assessment ofAntidiabetic, Hepatoprotective, and Analgesic Effects of Quinazolinone Derivative, (E)-1-Benzoyl-3-((4- (Dimethylamino) Benzylidene) Amino)-2-(4-(Dimethylamino) Phenyl)-2,3 dihydroquinazoline-4(1h)-one, in Diabetes induced Mice Model

PONE-D-25-32082R2

Dear Dr. Nahla,

We’re pleased to inform you that your manuscript has been judged scientifically suitable for publication and will be formally accepted for publication once it meets all outstanding technical requirements.

Kind regards,

Oluwafemi Adeleke Ojo, Ph.D

Academic Editor

PLOS ONE

Additional Editor Comments (optional):

No further comments

Reviewers' comments:

Reviewer's Responses to Questions

**Comments to the Author**

Reviewer #1: All comments have been addressed

2. Is the manuscript technically sound, and do the data support the conclusions?

Reviewer #1: Yes

3. Has the statistical analysis been performed appropriately and rigorously?

Reviewer #1: Yes

4. Have the authors made all data underlying the findings in their manuscript fully available?

Reviewer #1: Yes

5. Is the manuscript presented in an intelligible fashion and written in standard English?

Reviewer #1: Yes

Reviewer #1: The authors have thoroughly revised the manuscript in accordance with the requested guidelines and feedback. All necessary modifications have been implemented to address the concerns and suggestions raised during the review process. Specific attention was given to refining the clarity, coherence, and overall structure of the content to enhance its readability and academic rigor. The revised version reflects a careful consideration of the reviewers’ comments, with appropriate adjustments made to the methodology, data interpretation, and presentation of results where applicable. Additionally, grammatical and typographical errors have been corrected to ensure a polished and professional final submission. The authors are confident that the updated manuscript now meets the expected standards and requirements, and they appreciate the constructive input that contributed to its improvement. A detailed response to each reviewer comment has also been provided to demonstrate how the revisions align with the feedback received. The manuscript is now resubmitted for further evaluation.

what does this mean?). If published, this will include your full peer review and any attached files.

**Do you want your identity to be public for this peer review?** For information about this choice, including consent withdrawal, please see our Privacy Policy

Reviewer #1: **Yes:** Akingbolabo Daniel Ogunlakin

---

## [Editor Report · Acceptance letter]

PONE-D-25-32082R2

PLOS One

Dear Dr. Abu Nahla,

I'm pleased to inform you that your manuscript has been deemed suitable for publication in PLOS One. Congratulations! Your manuscript is now being handed over to our production team.

Kind regards,

on behalf of

Dr. Oluwafemi Adeleke Ojo

Academic Editor

PLOS One